# Pancreatic Tumorigenesis: Oncogenic KRAS and the Vulnerability of the Pancreas to Obesity

**DOI:** 10.3390/cancers13040778

**Published:** 2021-02-13

**Authors:** Yongde Luo, Xiaokun Li, Jianjia Ma, James L. Abbruzzese, Weiqin Lu

**Affiliations:** 1The First Affiliated Hospital & School of Pharmaceutical Sciences, Wenzhou Medical University, Wenzhou 325035, Zhejiang, China; xiaokunki@wmu.edu.cn; 2Department of Medicine, Stony Brook University, Stony Brook, NY 11794, USA; Jianjia.Ma@stonybrookmedicine.edu; 3Division of Medical Oncology, Department of Medicine, Duke Cancer Institute, Duke University, Durham, NC 27710, USA; james.abbruzzese@duke.edu

**Keywords:** KRAS, obesity, FGF21, PDAC, inflammation

## Abstract

**Simple Summary:**

Pancreatic cancer is a devastating disease with a poor survival rate, and oncogenic mutant KRAS is a major driver of its initiation and progression; however, effective strategies/drugs targeting major forms of mutant KRAS have not been forthcoming. Of note, obesity is known to worsen mutant KRAS-mediated pathologies, leading to PDAC with high penetrance; however, the mechanistic link between obesity and pancreatic cancer remains elusive. The recent discovery of FGF21 as an anti-obesity and anti-inflammation factor and as a downstream target of KRAS has shed new light on the problem.

**Abstract:**

Pancreatic ductal adenocarcinoma (PDAC) is one of the most lethal malignancies and *KRAS* (Kirsten rat sarcoma 2 viral oncogene homolog) mutations have been considered a critical driver of PDAC initiation and progression. However, the effects of mutant KRAS alone do not recapitulate the full spectrum of pancreatic pathologies associated with PDAC development in adults. Historically, mutant KRAS was regarded as constitutively active; however, recent studies have shown that endogenous levels of mutant KRAS are not constitutively fully active and its activity is still subject to up-regulation by upstream stimuli. Obesity is a metabolic disease that induces a chronic, low-grade inflammation called meta-inflammation and has long been recognized clinically as a major modifiable risk factor for pancreatic cancer. It has been shown in different animal models that obesogenic high-fat diet (HFD) and pancreatic inflammation promote the rapid development of mutant KRAS-mediated PDAC with high penetrance. However, it is not clear why the pancreas with endogenous levels of mutant KRAS is vulnerable to chronic HFD and inflammatory challenges. Recently, the discovery of fibroblast growth factor 21 (FGF21) as a novel anti-obesity and anti-inflammatory factor and as a downstream target of mutant KRAS has shed new light on this problem. This review is intended to provide an update on our knowledge of the vulnerability of the pancreas to KRAS-mediated invasive PDAC in the context of challenges engendered by obesity and associated inflammation.

## 1. Introduction

PDAC is the major histologic subset of pancreatic cancer, accounting for more than 90% of all pancreatic malignancies. Despite advances in medical knowledge that has improved the survival rate of many types of aggressive cancer, PDAC remains one of the deadliest malignancies with an incidence/mortality ratio of as high as 94% [1,2] and a five-year survival rate of about 9% [3]. It is predicted that within a decade PDAC will rise to the second-leading cause of cancer-related deaths in the U.S. [4,5]. Even more threatening, there are currently no highly effective standard therapies for the large majority of unresectable PDAC patients. The development of novel preventive and therapeutic approaches for PDAC remains an unmet medical need.

Many factors contribute to this dire situation, with both genetic and environmental risk factors being critically at play. More than 95% of PDAC patients harbor oncogenic *KRAS* (Kirsten rat sarcoma 2 viral oncogene homolog) mutations, predominantly at the Gly12, Gly13, and Gln61 positions, with *KRAS^G12D^* and *KRAS^G12V^* being the most common [6,7]. Mutant KRAS has been thought to be constitutively active and drive unintended and overactive signaling inside the cells even in the absence of external stimuli, resulting in uncontrolled cell proliferation, enhanced survival, resistance to drug treatment, and metabolic reprogramming such as adaptation to anaerobic glycolysis, leading to chronic tissue damage, persistent inflammation, and neoplastic transformation [6,8]. Although it has been more than three decades since the discovery that the RAS family is capable of driving cellular transformation when mutated, and there has been extensive research invested in various inhibitors targeting mutant RAS and associated downstream signaling pathways, its cellular localization, post-translational modifications, and associated metabolic changes, to date, no effective anti-mutant RAS^G12D/V^ inhibitors have successfully made their way to the clinic. 

Obesity and pancreatitis (pancreatic inflammation) are known clinical risk factors that are positively associated with PDAC development. Of note, obesity itself is also known to cause not only chronic metabolic abnormalities but also meta-inflammation and associated damage to the pancreatic tissue. Over the past thirty years, the prevalence of obesity in the U.S. has been increasing at an alarming rate. Obese adults made up less than 15% of the population in most States in 1990 and had grown to 23% by 2005 [9]. In 2018, this rate has rapidly increased to more than 30% according to the Centers for Disease Control and Prevention. With this trajectory, it is predicted that by 2030 nearly 1 in 2 adults will be obese [10]. Based on the National Institutes of Health (NIH) reports, overweight (body mass index (BMI) = 25.0–29.9 kg/m^2^) and obesity (BMI ≥ 30 kg/m^2^) are associated with increased risk for 13 types of cancer and can adversely affect cancer treatment and survival. As obesity is primarily a dysfunction of adipose tissues with systemic metabolic maladaptation playing a critical role, many studies have explored the metabolic and secretory functions of adipose tissues and the contribution of its chronic perturbation to neoplastic progression. Despite the clear epidemiological contribution of obesity to pancreatic cancer, the mechanistic link between obesity and pancreatic cancer remains largely unclear, and most of the current anti-obesity or adipose targeting approaches have proven to be ineffective for both prevention and treatment of PDAC. New mechanistic insights and the development of novel preventive and therapeutic strategies are critically needed.

In this review, we will focus on the interplay of obesity (e.g., adipose tissue dysfunction and meta-inflammation), oncogenic KRAS, and metabolic and inflammatory regulators, in particular FGF21, a novel anti-obesity and anti-inflammation factor, in the development of PDAC.

## 2. Roles of Obesity in the Development of PDAC 

Obesity is considered a medical condition characterized by the accumulation of excessive fat to an extent that causes metabolic derangement, inflammation, and organ dysfunction, as well as comorbidities such as insulin resistance, type 2 diabetes, hyperlipidemia, and cardiovascular diseases. Fat can also accumulate in organs such as the liver and pancreas contributing to cancer development. The pathognomonic sign of obesity is the abnormal levels of fat deposition in adipose tissues, in particular subcutaneous and visceral adipose tissues. The chronic consumption of HFD is known as a primary dietary factor in promoting obesity and studies have shown a positive association between obesity/HFD consumption and pancreatic cancer [11,12,13,14,15], and high BMIs were associated with an increased risk of death for PDAC patients [16,17]. A pooled analysis of 14 cohort studies assessing anthropometric factors and pancreatic cancer risk showed that patients who were overweight or obese were about 1.5 times as likely to develop pancreatic cancer as patients with normal range BMIs [18]. Similarly, results of a large NIH Diet and Health Study cohort showed a significant positive association between overweight and obesity in early, middle, and older ages and subsequent pancreatic cancer. Individuals with the longest duration of overweight and diabetes were at the greatest risk of developing pancreatic cancer [19]. The strong positive association between obesity and PDAC suggests a role for deranged adipose tissue and altered lipid metabolism in promoting pancreatic cancer initiation and progression. It is anticipated that efforts to uncover the molecular mechanisms linking obesity, adipose dysfunction, and PDAC will contribute to the discovery of novel targets for effective prevention or treatment of obesity-related pancreatic cancer.

The molecular mechanisms linking PDAC and obesity are complex and not fully understood. Peripheral, peri-pancreatic adipose tissues and tumor-residing adipocytes or adipose-like cells are each considered to play a role. Adipose tissues or adipocytes under obesity conditions demonstrate altered profiles of systemic and local levels of signaling molecules, including insulin, insulin-like growth factor 1 (IGF1), adipokines, cytokines, chemokines, growth factors, sex hormones, and nutrient substrate molecules, forming a pro-tumorigenic milieu [20,21]. Hypoxia and dysregulation of glucose/lipid metabolism in obese conditions, secondary to abnormal fat accumulation, are also key promoters of chronic, low-grade inflammation in both the adipose tissue and the associated tumors [22]. Similarly, an adipose tissue dysfunctional pro-inflammatory microenvironment may be induced by interaction with local tumor cells and tumor-secreted factors, independent from obesity with both obesity-related and tumor-related mechanisms forming a dysfunctional metabolic milieu [23,24,25]. These mutually reinforcing microenvironmental interactions contribute to (1) derangement of local and systemic glucose, lipid, and energy metabolic homeostasis [26,27,28]; (2) altered secretory functions as well as the local and systemic release of pro-tumorigenic and pro-inflammatory elements, such as growth factors, cytokines, chemokines, hormones, nutritional substrates, and extracellular vesicles [29,30,31,32,33]; (3) induction of stress, such as the endoplasmic reticulum (ER) stress, oxidative stress, and stress-induced tissue damage [34]; (4) activation, expansion, and recruitment of inflammatory and immune cells [35]; (5) alteration of tumor immunity and immune evasion [36,37]; (6) the extracellular matrix (ECM) remodeling, stromal cell activation, accumulation of adipose stem/stromal cells, fibrosis, and desmoplasia [30,38]; (7) stimulation of tumor cell growth, tumor angiogenesis, and lymphangiogenesis [39,40]. Thus, these local and systemic alterations occurring as a result of chronic obesity create an environment fostering the initiation, progression, and invasiveness of cancer, including PDAC. 

Expansion and inflammation of fat depots are considered responsible for the local and systemic release of excessive free fatty acids (FFAs), glycerol, and diacylglycerol. Excessive FFAs tend to be stored in non-adipose organs such as the pancreas and others (liver, muscle, heart, and kidney), promoting fatty pancreas disease, insulin resistance, hyperinsulinemia, and diabetes accompanied by increased levels of insulin and IGF1 [41,42,43,44]. Chronically enhanced insulin/IGF1 signaling has been found to promote pancreatic cancer cell proliferation and survival through both the insulin-like growth factor 1 receptor (IGF1R) and insulin receptor-mediated pathways, which activate the downstream phosphatidylinositol 3-kinase (PI3K), mammalian target of rapamycin (mTOR), and the mitogen-activated protein kinase (MAPK) signal pathways, as well as downregulating the tumor suppressor PTEN [45,46,47]. Experimentally, chronic consumption of a HFD is often used to induce obesity, diabetes, fatty tissue diseases, and meta-inflammation in several tissues, and to promote both chemically-induced and oncogenic KRAS-mediated pancreatic carcinogenesis [48,49,50,51,52,53,54,55]. *Kras^G12D/+^* expressing mice fed a high fat, high-calorie diet developed early pancreatic neoplasia with signs of obesity, hyperinsulinemia, hyperglycemia, hyperleptinemia, hyperlipidemia, and high levels of IGF1, resulting in significant activation of pancreatic stellate cells, recruitment of inflammatory cells, and ECM remodeling [51]. In marked contrast, *Kras^G12D/+^* expressing mice fed a calorie-restricted diet showed a reduction in serum IGF1, inhibition of AKT/mTOR signal pathways, suppression of pancreatic desmoplasia, and reduced progression to PDAC [56,57]. The Zucker diabetic and obese rats chronically fed a HFD developed signs of the fatty pancreas, pancreatic injury, and fibrosis with significantly elevated levels of FFAs, triglycerides, insulin, and monocyte chemoattractant protein-1. Consistent with these observations, clinical studies in PDAC patients showed a significant correlation between increased serum IGF1 concentration and poorer survival [58].

White adipose tissue also secretes other so-called adipokine soluble factors, notably leptin and adiponectin, among others. Leptin functions by binding and activating the leptin receptor (Ob-R), a member of the class I cytokine receptor family, and through cross-talk with the Notch receptor pathways and signals satiety to the hypothalamus to regulate dietary intake, energy expenditure, and fat storage [59]. Obesity is associated with leptin resistance despite increased leptin levels [60]. Case-control studies showed that high plasma leptin concentrations were associated with an increased risk of developing PDAC [61,62]. Obese conditions induce hypoxia and activate hypoxia-inducible factor 1-alpha (HIF1a), which was shown to directly activate Ob-R in pancreatic cancer cells [63]. A growing body of experimental evidence suggests that increased leptin signaling promotes pancreatic cancer progression, metastasis, and chemoresistance [64,65]. On the other hand, low serum concentrations of adiponectin, another unique adipose-derived adipokine, was found inversely associated with increased visceral fat in overweight and obese subjects and was associated with worse outcomes for PDAC patients [66,67]. *Kras^G12D/+^* expressing mice exposed to intermittent and chronic calorie-restricted diets showed a reduced incidence of pancreatic intraepithelial neoplasia (PanIN) and delayed PDAC progression accompanied by increased serum adiponectin and decreased serum leptin levels [57]. 

Pathological expansion of visceral adiposity and accumulation of fat in obese conditions are also known to evoke pro-inflammatory or inflammatory cellular changes characterized by a pattern of prominent infiltration of macrophages, T helper type 1 (Th1) cells, neutrophils, monocytes, and natural killer cells, as well as production and secretion of a plethora of pro-inflammatory cytokines, such as interleukin 6 (IL6), tumor necrosis factor alpha (TNFα), and interleukin 1 alpha (IL1α). These changes can occur in the peripheral adipose tissues, peri-pancreatic fat, and the pancreas or pancreatic tumor foci, contributing to PDAC development and invasiveness [22,68,69,70]. In turn, increased IL6, TNFα, and IL1α levels were showed to significantly activate c-Jun N-terminal kinases, nuclear factor-κB (NFκB), signal transducer and activator of transcription 3, Janus kinase 2, and MAPK signaling pathways, promoting inflammation, stromal cell activation, and pancreatic cancer cell growth. Deletion of tumor necrosis factor receptor 1 or IL6 in *Kras^G12D/+^*-expressing mice significantly reduced HFD- or inflammation-promoted tissue damage, fibrosis, cancer cell proliferation, and PanIN development [50,71]. Furthermore, these pro-inflammatory cytokines also regulate the release of some adipokines and other cytokines, worsening the adipose tissue associated pro-inflammatory state, altering tumor-stroma interactions further activating inflammatory signal pathways thus contributing to the development of invasive PDAC [72].

Pancreas-residing or -infiltrating adipocytes or adipose-like cells such as stromal cells are as important for maintaining pancreatic structure and function as the peripheral and peri-pancreatic adipose tissues. Under obese and pro-tumorigenic conditions, the intra-pancreatic fat-laden cells (steatosis) interact with tumor cells and non-fat stromal or precursor cells, producing an abnormal pro-tumorigenic, pro-inflammatory, and pro-fibrotic symbiosis. With mechanisms similar to the aforementioned for peripheral and peri-pancreatic adipose tissues, this symbiosis promotes the abnormal local release of cytokines, activation of pancreatic stromal cells, infiltration of myofibroblast-like cells and immune cells, accumulation of ECM, angiogenesis, and fibrosis, resulting in pancreatic desmoplasia which, in turn, promotes growth, progression, and treatment resistance of PDAC [22,73,74,75]. Pancreatic steatosis (fatty pancreas) and the infiltration of the peri-pancreatic fat are associated with poor clinical outcomes in human PDAC patients [76,77,78]. In KPC (*Ptf1^Cre^*;*KRAS^LSL-G12D/+^*;*Trp53^LSL-R172H/+^*) mice fed a HFD, the rapid development of invasive PDAC and resistance to chemotherapy were accompanied by a significant increase in the number and size of intra-tumoral fat cells, impaired vascular perfusion, secretion of IL1β, subsequent recruitment of tumor-associated neutrophils, macrophages, lymphocytes, and activation of pancreatic stromal cells [30,68]. In summary, activation of an inflammatory response in the intra- or peri-pancreatic adipose tissues or fat-like cells in the context of obesity induces inflammation and fibrosis in the neighboring pancreas and the cellular precursors to pancreatic cancer, promoting progression to aggressive PDAC and drug resistance.

These observations prompted a number of clinical studies with various inhibitors targeting adipokines, mediators of inflammation and associated secretory factors, and abnormal lipid metabolism for the prevention and treatment of PDAC. However, thus far these inhibitors have shown no notable clinical benefits to PDAC patients. Trials with Cixutumab/Ganitumab, Etanercept, Anakinral, and Tocilizumab that target IGF1R, TNFα, IL1 receptor, and IL6 receptor, respectively, alone or in combination with other chemotherapy agents, have produced disappointing results with no improvements in clinically meaningful parameters such as time to disease progression, median progression-free survival, disease control rate, or overall survival [79,80,81]. 

Studies also suggested that broadly targeting the abnormal lipid metabolic profile in PDAC patients could be a viable therapeutic approach. Use of statins that inhibit HMG-CoA reductase mediated cholesterol synthesis has been associated with decreased human pancreatic cancer risk and reduction of pancreatic cancer development and cancer cell growth in pre-clinical studies [82,83]. However, phase 2 randomized clinical trials with simvastatin in advanced PDAC patients showed no improvement in clinical outcomes [84]. Metformin is a widely used treatment agent for type 2 diabetes associated with obesity and the metabolic syndrome thought to function by targeting the mTOR and AMPK pathways. Encouragingly, studies in a pancreatic cancer mouse model with diabetes/obesity showed inhibitory effects of metformin on tumor progression [85]. However, multiple human clinical trials with metformin failed to show advantages in clinical outcomes even in combination with other standard chemotherapies [86,87,88]. In sum, clinical outcomes with inhibitors targeting obesity-associated abnormal lipid metabolism, soluble factors such as cytokines and growth factors, and inflammatory pathways in PDAC patients have so far failed to achieve significant clinical impact despite the promising leads generated by the mechanistic and preclinical insights described above. A deeper understanding of the mechanistic role of mutant KRAS and physiological and biochemical changes precipitated by obesity, diets high in fat, and the associated inflammatory microenvironment is needed to develop novel and effective preventive and therapeutic strategies for PDAC. 

## 3. Roles of Oncogenic KRAS in the Development of PDAC 

The *RAS* family member genes, including *HRAS, NRAS,* and *KRAS*, are the most frequently mutated oncogenes found in as high as 30% of all human cancers, among which mutations in KRAS constitute 86% of all RAS mutations [89,90]. Pancreatic cancer-associated KRAS gene mutations include the missense mutations that result in single amino acid substitutions primarily at codon G12 (98%) with lower frequencies at G13 or Q61. G12D is the predominant mutation accounting for 51% of all mutations at codon G12 (COSMIC, http://cancer.sanger.ac.uk/cancergenome/projects/cosmic/). Despite the critical importance of oncogenic KRAS mutations in PDAC patients, direct molecular targeting of oncogenic KRAS has been thus far problematic. Except for the recent successful development of the specific inhibitors for KRAS^G12C^, which is found in only 3% of PDAC patients [91,92,93,94,95,96,97], no specific inhibitors for KRAS^G12D/V^ have been reported to date. Thus, new preventive and therapeutic strategies directly or indirectly targeting KRAS^G12D/V^ are critically important for dealing with this devastating disease. 

The *KRAS* gene is a proto-oncogene that encodes a small GTPase, which functions as a binary molecular switch. When bound with guanosine diphosphate (GDP), KRAS is inactivated, which denotes the so-called “off” state, while when bound with guanosine triphosphate (GTP) in response to an external signal or stimulus, KRAS is activated, which signifies a switch to the so-called “on” state [98,99]. The KRAS-GDP to KRAS-GTP conversion is regulated by guanine nucleotide exchange factors (GEFs), such as the Son of Sevenless (SOS), that promote nucleotide exchange and formation of active KRAS-GTP [100]. The activation of KRAS promotes various downstream signaling pathways, such as the MAPK pathway, the PI3K pathway, and the Ral-GEFs pathway, leading to an array of cellular responses, including cell proliferation, growth, survival, migration, and metabolic reprogramming [101,102]. The switch from KRAS-GTP to KRAS-GDP is facilitated by GTPase-activating proteins (GAPs) [103]. KRAS molecules have intrinsic GTPase activity and upon binding with GAPs the rate of GTP hydrolysis is accelerated, generating GDP-bound inactive KRAS, the predominant state under normal physiological conditions [104,105,106]. Thus, the activation of wild-type KRAS is transient (Figure 1A).

Previous studies showed that mutations in KRAS disable its interaction with GAPs, and thus impair its ability to return to the “off” state. For this reason, over the past thirty years, it has been assumed that mutant KRAS is locked in a permanent “on” and constitutively active state, which is capable of driving uncontrolled cell division, growth, survival, invasion, and metabolic maladaptation characteristic of cellular transformation without external stimuli [98,107,108]. Many mouse models have been created to recapitulate the tumorigenic processes driven by oncogenic mutant KRAS, in particular the development of PDAC. The widely used Pdx1-KC (*Pdx-1^Cre^*;*Kras^LSL-G12D^*) mouse model was generated using a Cre-LoxP technology by crossing the *Pdx1^Cre^* mice with *KRAS^LSL-G12D/+^* mice [109]. Following activation of Cre recombinase, the Pdx1-KC mice simulate an endogenous level of KRAS^G12D^ in pancreatic progenitor cells. The constitutive, endogenous levels of KRAS^G12D^ expression from the embryonic stage induced a full spectrum of pancreatic neoplastic pathologies, including PanIN lesions and PDAC; however, the incidence of PDAC development was only 7% within a year, far less than what was predicted [109]. Substantial evidence supports that acinar cells are the cell of origin for PDAC [110,111,112]. Given that PDAC is likely initiated during adulthood by somatic mutations in *KRAS* in acinar cells rather than during embryonic development, a mouse model that allowed tetracycline-controlled expression of endogenous levels of KRAS^G12V^ specifically in pancreatic acinar cells in the adult mice was developed [113,114]. This model demonstrated that adult pancreatic acinar cells are rather refractory to transformation by KRAS^G12V^ alone and no PanIN lesions or PDAC foci were observed. Similarly, mouse models expressing oncogenic KRAS at endogenous levels showed that the number of transformed cells was only a small fraction of those expressing oncogenic KRAS [115]. These observations suggest that mutant KRAS alone is insufficient to drive full-blown PDAC and other factors or a second hit is required. This conclusion is supported by studies showing that *KRAS* mutations are also frequently detected in healthy individuals [116,117]. 

Comparison of KRAS activity from pancreatic tissue samples expressing one copy of *Kras^G12D^* and one copy of wild-type *Kras* in pancreatic acinar cells to those expressing wild-type KRAS showed that the level of GTP-loaded RAS in cells carrying one copy of *Kras^G12D^* was far less than expected if KRAS^G12D^ was constitutively GTP occupied [118,119,120]. Specifically, mutant KRAS activity was less than 2%, which is in marked contrast to what was expected when 50% of the total KRAS protein was mutated [120]. Furthermore, the expression of endogenous levels of mutant KRAS failed to significantly activate its downstream signaling pathways and to induce notable pathological changes [113,115]. Direct evaluation of GTP occupation of KRAS indicated that stimulants such as EGF could significantly increase GTP-bound active KRAS for a prolonged time in cells bearing endogenous levels of mutant KRAS [115]. Expression of a dominant-negative GEF reduced RAS activity in cells expressing mutant RAS [121], suggesting that, rather than being constitutively active, mutant RAS activity is still subject to the regulations by GEF as seen with the wild-type KRAS. Upon interaction with GEFs, mutant KRAS exhibited prolonged activation relative to wild-type KRAS [120]. These observations suggest that mutant KRAS is not fully active as previously thought and can be further enhanced by upstream stimuli with prolonged activation. Overexpression of mutant KRAS by strong promoters was highly effective in transforming cells [122], which was attributed to the high levels of KRAS activity. This is in contrast to endogenous levels of mutant KRAS, which appears to exhibit lower levels of KRAS activity and is ineffective in transforming cells (Figure 1B). Unlike what had been assumed to be the pervasive roles of mutant KRAS in pancreatic oncogenesis, these data suggest that an endogenous level of oncogenic KRAS is far less than fully active and has limited capacity to drive the development of PDAC and therefore, a second hit is required for enhancing oncogenic KRAS activity that can drive full-blown PDAC. In this regard, obesity and inflammation are considered among the most predominant clinical risk factors and are likely to synergize with mutant KRAS to drive pancreatic tumorigenesis leading to invasive PDAC, in which oncogenic KRAS is known to be hyperactivated (Figure 1C). These studies suggest that the regulation of mutant KRAS activity is complex and our understanding of these mechanisms is currently incomplete.

## 4. Interplay between Obesogenic HFD, Inflammation, and Oncogenic KRAS in PDAC Development

Although several genetically engineered mouse models mimicking endogenous levels of mutant KRAS expression in pancreatic acinar cells demonstrated that oncogenic KRAS alone is marginally effective at inducing full-blown PDAC, the combination of oncogenic KRAS and chronic HFD consumption led to extensive pancreatic inflammation, fibrosis, high-grade PanIN lesions, and invasive PDAC [49,50,51,52,53]. In addition, KRAS^G12D^ mice fed a HFD exhibited significantly increased levels of glycolytic enzymes including hexokinase II and lactate dehydrogenase A compared to mice fed a control diet, indicating that a key metabolic effect of mutant KRAS (e.g., enhancement of anaerobic glycolysis) is magnified in the setting of obesogenic HFD challenge [123]. These observations indicate that mutant KRAS renders mice more susceptible to adverse insults of chronic HFD, leading to the enhancement of oncogenic KRAS-mediated development of invasive PDAC. 

Like obesity, chronic pancreatitis is also a known risk factor for PDAC development. When subjected to caerulein (an agonist of cholecystokinin (CCK))-induced pancreatitis, adult mice that express endogenous levels of KRAS^G12V^ in the pancreatic acinar cell compartment developed extensive PanIN lesions and PDAC with high penetrance [113,114]. Similarly, overexpression of cyclooxygenase-2 (COX-2) in pancreatic acinar cells of fElas^CreERT^-KC mice that express endogenous levels of KRAS^G12D^ induced RAS hyperactivation and associated extensive inflammation, fibrosis, and rapid development of high-grade PanIN lesions [115]. Consistent with these observations, constitutive activation of NFκB by overexpressing IKK2 (inhibitor kappa B kinase 2) in pancreatic acinar cells of fElas^CreERT^-KC mice led to the rapid development of neoplastic lesions [115]. These results are also in line with other studies showing that although the insulin-expressing endocrine cells were refractory to oncogenic *KRAS*-induced transformation, chronic pancreatic inflammation could cooperate with oncogenic KRAS to alter the endocrine phenotype of these cells, leading to pancreatic neoplastic progression in cells generally refractory to neoplastic transformation [124]. These experimental studies highlight the critical role of inflammation in promoting oncogenic KRAS-mediated development of pancreatic neoplastic lesions, supporting an important synergy between mutant KRAS and inflammation in driving PDAC development [113,114,124,125,126]. Furthermore, inhibition of inflammation by ablating COX-2 or IKK2 curbed the synergy between inflammation and oncogenic KRAS, leading to significant suppression of inflammation and fibrosis, PanIN lesions, and PDAC development [115]. 

Notably, studies showed that the inflamed pancreata from the aforementioned mice suffering from obesity and extensive inflammation exhibited a significant increase in KRAS activity (Figure 1C). In addition, when treated with inflammatory stimulants, such as CCK, prostaglandin E2, or lipopolysaccharide, the amount of GTP-bound mutant KRAS is increased and prolonged compared to that of wild-type KRAS [115]. In marked contrast, inhibition of HFD challenge by the administration of recombinant FGF21 (see below) or suppression of inflammatory stimuli by ablation of COX-2 or IKK2 curbed RAS activity with notable attenuation of PanIN lesions and malignant progression to PDAC [49,53,115,123]. These studies suggest that the hyperactivation of mutant KRAS by obesogenic HFD and inflammation, rather than the simple presence of an endogenous level of mutant KRAS protein alone, is critical for the development of PDAC [118,119]. Although oncogenic mutation in KRAS is a necessary genetic transforming event and confers susceptibility to additional challenges, obesogenic HFD challenge or inflammatory insults serve as the second hit to enhance KRAS oncogenic capacity, driving pancreatic tumorigenesis and the eventual development of invasive PDAC. Mechanistically, the prolonged activation of mutant KRAS led to the activation of NFκB, COX-2, and others, creating a positive feedback loop to sustain mutant KRAS hyperactivity, leading to the rapid development of invasive PDAC [49,115]. This appears to be true in the early stage of PDAC development since the ablation of COX-2 or IKK2, as well as inhibition of COX-2 by celecoxib, suppressed mutant KRAS-mediated PDAC development [49,115]. However, interestingly, in vitro studies showed that neither the presence of oncogenic *KRAS* nor the level of mutant KRAS activity positively correlated with COX-2 protein levels in pancreatic cancer cells [120,127,128], suggesting that COX-2 overexpression and resultant inflammation might be a critical early event in KRAS hyperactivation during PDAC development. When the activity of mutant KRAS reaches a certain level or a threshold, external stimuli, such as COX-2, may no longer be required for the development of the late-stage PDAC. This may in part explain the failure of the inhibitors of obesogenic or inflammatory factors as aforementioned in deterring PDAC development in patients even in combination with other chemotherapy agents that do not target mutant KRAS. Recently, it has also been shown that there are two distinct inflammation states associated with the macrophages and Th cells in pancreatic cell microenvironment during PDAC development, the so-called Th1-M1 state and the Th2-M2 state, which correspond to an inflammatory and an anti-inflammatory phenotype, respectively. Evidence suggests that the Th1-M1 state may predominate the early ADM and PanIN stages, while the PDAC is mostly associated with the anti-inflammatory Th2-M2 state. This may also explain why anti-inflammatory agents have little effect on diminishing PDAC progression while possibly having a role in preventing the precursory ADM and early PanIN lesions [72,129,130]. Therefore, a detailed understanding of the vulnerability of the pancreas to exogenous risk factors during PDAC development and the underlying mechanisms of the interplay between oncogenic KRAS and the environmental risk factors, including obesity, chronic HFD, and inflammation, are critical for the design of novel, effective targeting strategies for the prevention and therapy of pancreatic cancer. 

## 5. FGF21: A Potential Missing Link between Obesity/Inflammation and Mutant KRAS-mediated Pancreatic Tumorigenesis

As a member of the FGF family, FGF21 is unique in that it functions to maintain lipid and energy homeostasis under normal physiological conditions, promoting metabolic homeostasis rather than stimulating cell proliferation and growth as most FGF family members do. Pharmacological levels of FGF21 exert both anti-obesity and anti-inflammatory activities. As an endocrine hormone, FGF21 is produced in the liver in response to local liver perturbations as well as systemic metabolic derangements and acts primarily on white and brown adipose tissues and the hypothalamus where the canonical transmembrane fibroblast growth factor receptor 1 (FGFR1) kinase and a newly described accessory non-kinase membrane-bound co-receptor Klotho beta (KLB) complex is located to promote the correction of the deranged metabolic parameters and the maintenance of metabolic homeostasis. The serum levels of FGF21 are increased in the settings of hyperlipidemia, obesity, diabetes, and fatty liver [131,132,133]. Adipose tissues also produce FGF21 as an adipokine, which potentially acts in a paracrine and autocrine mode [134,135,136]. Administration of exogenous FGF21, FGF21 analogs, or FGFR1-KLB agonists promotes an array of metabolic benefits in mice that intercept obesity, type 2 diabetes, fatty liver disease, and hyperlipidemia [137,138,139,140,141]. It is important to note that adipose tissue is the primary as well as the ultimate (in terms of some of the central nervous system routes) target for major effects of FGF21 where its receptor complex FGFR1-KLB is predominantly located [138,142,143,144,145]. The pharmacologic impact of FGF21 on obese and inflamed adipose tissues, residential adipocytes, and the associated metabolic abnormalities and inflammatory damage in the context of obesity is expected to be more direct and significant than some of the known metabolic and inflammatory regulators, such as leptin, adiponectin, and IL6. 

A number of studies have revealed that FGF21 is expressed also at high levels at normal conditions in the pancreas, in particular the pancreatic acinar cells, which can be further induced in response to pancreatic perturbations, such as the fatty pancreas, pancreatitis, and pancreatic injury [53,146,147]. FGF21 also functions as a secretagogue to promote pancreatic exocrine function and maintain acinar cell proteostasis, which if uncontrolled could cause pancreatitis. Loss of FGF21 results in abnormal accumulation of zymogen granules that induces pancreatic ER stress (Table 1) [148]. Similarly, mice lacking FGF21 developed significant islet hyperplasia and periductal lymphocytic inflammation with elevated expression of cytokines such as TNFα, interferon gamma, and IL1β when fed an obesogenic diet [147], while liver-specific overexpression of FGF21 reduces pancreatic β-cell apoptosis in *db/db* mice (Table 1) [149]. Consistent with the beneficial effects of FGF21 on obesity and fatty liver disease, pharmacological FGF21 protects the pancreas from gluco-lipotoxicity and cytokine-induced damage, pancreatitis, and fibrosis both in the islet and acinar cell compartments, promoting both endocrine and exocrine functions of the pancreas in various mouse models, such as the Streptozotocin (STZ)-induced type 1 diabetes, *ob/ob*, *db/db*, and diet-induced obese (DIO) models [150,151,152,153,154,155]. Deficiency of FGF21 under pancreatitis induced by caerulein, thapsigargin, or mechanical stress worsened pancreatitis and precipitated overt organ damage, while overexpression of FGF21 reversed these effects (Table 1) [156]. FGF21 treatment in various mouse models of acute and chronic pancreatitis induced by caerulein, alcohol consumption, or endoscopic retrograde cholangiopancreatography radiocontrast dye prevented the onset of pancreatitis and alleviated symptoms, while acinar cell-specific deficiency or silencing of FGF21 or its co-receptor KLB blocked these beneficial effects [156,157]. These data suggest that FGF21 deficiency is necessary for pancreatitis and that FGF21 has the potential for treating pancreatitis. 

Both the pancreas and the liver are metabolic organs and are subject to the development of a spectrum of the fatty pancreas and fatty liver diseases under conditions of chronic caloric excess. The protective anti-obesity (including the obesity complications), anti-diabetes, and anti-inflammation effects of pharmacological FGF21 have been extensively studied in models of fatty liver disease including hepatosteatosis, non-alcoholic steatohepatitis (NASH), and in malignancies such as hepatocellular carcinoma [162,163,164,165,166,167]. In clinical trials with patients who were obese or diabetic with a predisposition to fatty liver complications including NASH, long-acting FGF21 analogs improved serum and hepatic metabolic parameters and hepatic fibrosis markers, suggesting that FGF21 is a promising therapy for obesity- and meta-inflammation-associated liver diseases [140,168,169]. Based on the resemblance in the pathological spectrums between fatty liver disease and fatty pancreas disease, it can be extrapolated that FGF21 would similarly exert an inhibitory, protective effect on the fatty pancreas and inflammation-associated PDAC development.

Furthermore, in several mouse models of mutant KRAS^G12D/+^-mediated PDAC as well as in patient samples, the expression of FGF21 is negatively associated with the expression of mutant KRAS^G12D/+^, suggesting that FGF21 is potentially a downstream target of oncogenic KRAS [53]. Silencing the expression of FGF21 by oncogenic KRAS is proposed to create a vulnerability in acinar cells to metabolic and inflammatory challenges, leading to a loss of a cell-autonomous defense system against metabolic and inflammatory insults that in concert with oncogenic KRAS promotes neoplastic transformation and accelerates the transition of low-grade PanIN lesions to high-grade lesions and invasive PDAC (Figure 2). Administration of pharmacological FGF21 compensates for the loss of pancreatic FGF21 induced by mutant KRAS^G12D^ in the context of an obesogenic HFD, leading to significant suppression of PanIN lesions, inhibition of invasive PDAC development, and reduction in liver metastasis [53]. Positive physiologic effects were also observed including inhibition of weight gain, pancreatic local and systemic inflammation, and pancreatic fibrosis. Taken together, FGF21 functions, at least in part, to offset the vulnerability that is incurred by oncogenic KRAS and hijacked by obesity and other inflammatory states. FGF21 may represent a promising next-generation preventive and therapeutic strategy that warrants comprehensive testing in human clinical trials of obesity-associated pancreatic cancer, pancreatitis, and other inflammation-related disease states such as NASH and hepatocellular cancer. 

## 6. Concluding Remarks and Future Perspectives

PDAC is currently one of the deadliest cancers in part owing to the lack of effective preventive and therapeutic approaches. Point mutations in the KRAS proto-oncogene, in particular G12D and G12V, are the dominant drivers of the initiation and progression of PDAC. Yet, inhibitors specifically targeting the major KRAS oncoproteins relevant to pancreatic cancer have not been successfully developed. Of note, mounting evidence suggests that mutant KRAS alone is insufficient to drive PDAC development in adults and environmental risk factors such as obesity and pancreatic inflammation are found to synergize with mutant KRAS to induce oncogenic KRAS hyperactivation, promoting extensive inflammation, fibrosis, and the rapid development of PanIN lesions and PDAC with high penetrance. The recent discovery of the potent metabolic regulator FGF21 with a combined activity of anti-obesity and anti-inflammation and as a downstream target of oncogenic KRAS has raised new optimism for PDAC prevention and therapy. FGF21 analogs have been tested in obese or diabetic patients with NASH generating favorable outcomes and exhibiting inhibitory effects on hepatocellular carcinoma in association with fatty liver disease in animal models. Notably, pharmacological FGF21 has also been shown to inhibit not only KRAS^G12D^-mediated pancreatic inflammation, fibrosis, and PanIN lesions but also the development of malignant PDAC and liver metastasis even in the setting of obesogenic HFD challenge [53]. Future work should focus on further delineating the mechanisms of KRAS hyperactivation by obesity, chronic HFD, and inflammation, the critical factors required to sustain the prolonged KRAS activity in PDAC development, and how FGF21 analogs or pathway agonists should be utilized in a clinical setting for pancreatic cancer patients who also suffer from the effects of longstanding obesity and/or pancreatitis.

## Figures and Tables

**Figure 1 cancers-13-00778-f001:**
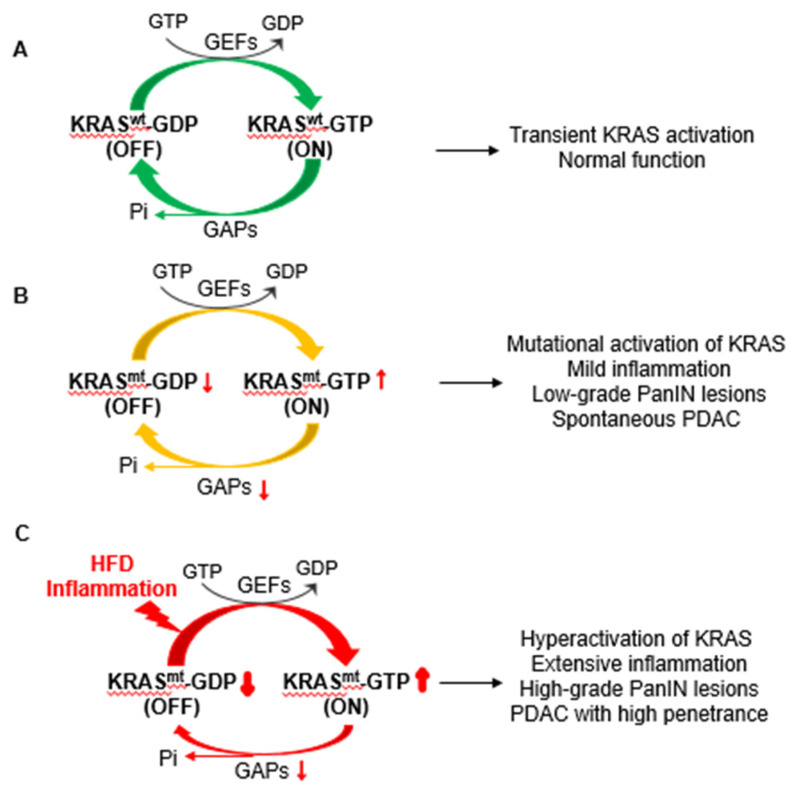
The activation of KRAS (Kirsten rat sarcoma 2 viral oncogene homolog) under different physio-pathological conditions. (**A**). Under physiological conditions, the activity of wild-type KRAS GTPase (KRAS^wt^) is tightly controlled by guanine nucleotide exchange factors (GEFs) and GTPase-activating proteins (GAPs), maintaining KRAS in an inactive KRAS^wt^-guanosine diphosphate (GDP) state unless an upstream external signal stimulates GEFs, which in turn promote the loading of guanosine triphosphate (GTP) in place of GDP to KRAS and thus switch the inactive KRAS^wt^-GDP state to active KRAS^wt^-GTP state. When interacting with GAPs, the intrinsic GTPase activity of KRAS^wt^ is greatly accelerated, which hydrolyzes the active KRAS^wt^-GTP to inactive KRAS^wt^-GDP, leading to transient activation of KRAS. (**B**). Oncogenic mutations in KRAS (KRAS^mt^) disrupt interactions with GAPs. However, these mutations neither alter the interactions with GEFs nor affect the intrinsic GTPase activity of KRAS and thus, KRAS^mt^ is still subject to GEF stimulation, potentially resulting in a prolonged KRAS^mt^-GTP state and delayed inactivation to KRAS^mt^-GDP. Such mutational activation of KRAS has been thought to drive mild inflammation, low-grade PanIN lesions, and spontaneous pancreatic ductal adenocarcinoma (PDAC) development when expressed at the embryonic stage or the early stage after birth. (**C**). Chronic high-fat diet (HFD) or pancreatic inflammation hyperactivates mutant KRAS, leading to extensive inflammation, high-grade PanIN lesions, and rapid PDAC development. GEF, guanine nucleotide exchange factor. GAP, GTPase activating protein. HFD, high-fat diet.

**Figure 2 cancers-13-00778-f002:**
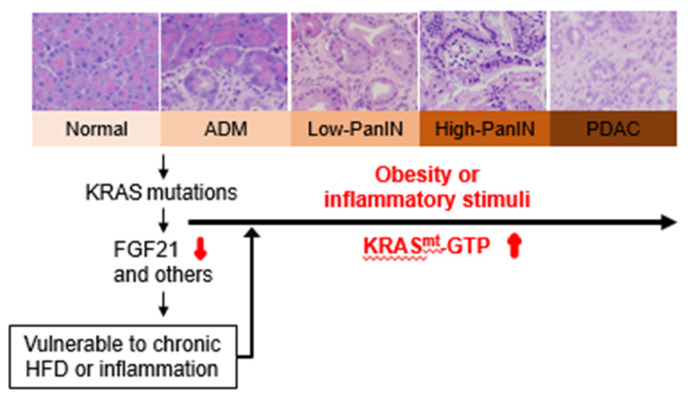
The mechanistic interplay of mutant KRAS and obesity/inflammation. Oncogenic KRAS not only directly or indirectly promotes cell proliferation, resistance to apoptosis, alteration of metabolism but also silences the expression of FGF21, a novel anti-obesity and anti-inflammation factor, leading to irreversible acinar-to-ductal metaplasia (ADM), fibrosis, and PanIN lesions. The loss of acinar cell FGF21 (and the alteration of other genes) by mutant KRAS promotes vulnerability in the acinar cell compartment to chronic HFD or inflammatory challenge, leading to KRAS hyperactivation, accentuated pancreatic inflammation, advanced PanIN lesions, and subsequent development of PDAC with high penetrance. Images were from the experiments of our laboratory.

**Table 1 cancers-13-00778-t001:** The effects of fibroblast growth factor 21 (FGF21) signaling in the pancreas.

FGF21-Related Models	Mouse Genotype	Phenotype or Effects	Reference
FGF21 overexpression	*ApoE^Cre^-FGF21Tg*	Protects acinar cells from caerulein, mechanical, or thapsigargin induced pancreatitis and stress damage.	[146]
		Protects acinar cells from caerulein, mechanical, or thapsigargin induced pancreatitis and stress damage.	[148]
		Reduces β-cell apoptosis in *db/db* mice.	[149]
		Mitigates acinar damage of *Mist1^−/−^* pancreas.	[156]
FGF21 knockout	Whole-body or germline *FGF21^−/−^*	Whole-body or germline *FGF21^−/−^*	[146]
		Exacerbates palmitate-induced pancreatic β-cell failure.	[149]
		Increases zymogen granules and susceptibility to ER stress in acinar cells.	[148]
		Induces insulin resistance, pancreatic islet hyperplasia, and dysfunction.	[158]
Acinar cell-specific *KLB^−/−^*	*Klb^Cela1-/-^*	Increases zymogen granules in acinar cells.	[148]
Injection of FGF21	WT *KRAS^LSL-G12D/+^*	Inhibits pancreatitis and fibrosis	[151,157]
		Compensates KRAS induced FGF21 loss to inhibit pancreatic inflammation, fibrosis, PanIN lesion, and PDAC development	[53]
Injection ofF GF21	Streptozotocin [STZ]-induced type 1 diabetes	Improves islet engraftment and insulin sensitivity.	[154]
		Prevents the increase in glycemia and lowers lipids.	[159]
Injection of FGF21	*ob/ob* and *DIO*	Improves glucose tolerance and insulin sensitivity, but has no direct effect on islet insulin secretion.	[155]
Injection of FGF21 or analogs	*db/db*	Improves islet survival and function, insulin sensitivity and glucose homeostasis.	[150,160]
		Increases pancreatic β-cell mass	[161]

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
