# Peer review of "Pancreatic Tumorigenesis: Oncogenic KRAS and the Vulnerability of the Pancreas to Obesity"

_cancers, 2021, doi:10.3390/cancers13040778_

Round 1

Reviewer 1 Report

Well written review.  

A few minor points:

  1. On p.4 it says,". However, thus far
    these inhibitors have produced notable clinical benefits to PDAC patients. Trials with
    Cixutumab/Ganitumab, Etanercept, Anakinral, and Tocilizumab that target IGF1R, TNFα,
    IL1R, and IL6R, respectively, alone or in combination with other chemotherapy agents,
    have produced disappointing results with no improvements in clinically meaningful parameters such as time to disease progression, median progression-free survival, disease
    control rate, or overall survival (79-81). "   

        If these compounds produced a notable clinical benefit then there should          clinical improvements, so maybe what you meant to say was that there            are NO notable benefits. 

      2. On p 7-8 it says, "These observations indicate
          that mice harboring mutant KRAS are vulnerable to chronic HFD                      consumption, leading to the enhancement of oncogenic KRAS-                       mediated development of invasive PDAC".  

          It almost sounds like you're saying that mutant Kras makes mice more            susceptible to consuming HFD.   So better if that's reworded.

Overall Remark:

I found this to be an informative review, especially the link between inflammation and upregulated oncogenic-Kras activity and then the possible downstream role for FGF21 in regulating inflammatory responses. 

It should be noted though that the current thinking seems to believe that two distinct inflammation states drive the early ADM/PanIN cycles(the so-called Th1-M1 states) while the actual PDAC is mostly driven by an anti-inflammatory state(Th2/M2).  This may explain why anti-inflammatory agents(eg COX2 inhibitors) have very little effect in diminishing PDAC progression while possible having a role in preventing the precursory ADM/PanIN stage.

Author Response

Response to the Comments and Suggestions from Reviewer #1

A few minor points:

  1. On p.4 it says,". However, thus far these inhibitors have produced notable clinical benefits to PDAC patients. Trials with Cixutumab/Ganitumab, Etanercept, Anakinral, and Tocilizumab that target IGF1R, TNFα, IL1R, and IL6R, respectively, alone or in combination with other chemotherapy agents, have produced disappointing results with no improvements in clinically meaningful parameters such as time to disease progression, median progression-free survival, disease control rate, or overall survival (79-81). "

        If these compounds produced a notable clinical benefit then there should          clinical improvements, so maybe what you meant to say was that there are NO notable benefits.

Answer: We have corrected this error accordingly.

  1. On p 7-8 it says, "These observations indicate that mice harboring mutant KRAS are vulnerable to chronic HFD consumption, leading to the enhancement of oncogenic KRAS-mediated development of invasive PDAC".

       It almost sounds like you're saying that mutant Kras makes mice more            susceptible to consuming HFD.   So better if that's reworded.

Answer: Thank you for the helpful comments. We have revised this sentence accordingly.

  1. Overall Remark:

I found this to be an informative review, especially the link between inflammation and upregulated oncogenic-Kras activity and then the possible downstream role for FGF21 in regulating inflammatory responses.

       It should be noted though that the current thinking seems to believe that two distinct inflammation states drive the early ADM/PanIN cycles (the so-called Th1-M1 states) while the actual PDAC is mostly driven by an anti-inflammatory state (Th2/M2).  This may explain why anti-inflammatory agents (eg COX2 inhibitors) have very little effect in diminishing PDAC progression while possible having a role in preventing the precursory ADM/PanIN stage.

Answer: This is a very helpful comment. We have added your points to the manuscript.

Reviewer 2 Report

Luo et al. have done a commendable job by nicely summarizing the synergistic roles of obesity and oncogenic KRAS as well as the FGF21 link in pancreatic tumorigenesis. This review calls for further research to better understand these links as well as for the development of novel therapeutic strategies keeping these links in mind. I have the following minor concerns:

  1. Please make sure all the abbreviations are expanded when used for the first time.
  2. Please note the the text in the figures have red underlines likely due PDF conversion from the text document.
  3. On page #8, there is some redundancy while mentioning the effects of ablation of COX-2 or IKK2 on PDAC development.
  4. Please make sure the source of images in cited in the legend of figure 2 as well as in the references.

Author Response

Response to the Comments and Suggestions from Reviewer #2

Luo et al. have done a commendable job by nicely summarizing the synergistic roles of obesity and oncogenic KRAS as well as the FGF21 link in pancreatic tumorigenesis. This review calls for further research to better understand these links as well as for the development of novel therapeutic strategies keeping these links in mind. I have the following minor concerns:

  1. Please make sure all the abbreviations are expanded when used for the first time.

Answer: Thank you for the helpful comments and we have revised the abbreviations throughout the text.

  1. Please note the text in the figures have red underlines likely due to PDF conversion from the text document.

Answer: we have revised the text used in the figures according to the suggestion.

  1. On page #8, there is some redundancy while mentioning the effects of ablation of COX-2 or IKK2 on PDAC development.

Answer: this is a very help suggestion, and we have revised the sections by deleting “Furthermore, inhibition of inflammation by ablating COX-2 or IKK2 curbed the synergy between inflammation and oncogenic KRAS, leading to significant suppression of inflammation and fibrosis, PanIN lesions, and PDAC development (115)”.

  1. Please make sure the source of images in cited in the legend of figure 2 as well as in the references.

Answer: The images were generated in our lab and we have indicated the source of images in the legend of Figure 2.

Reviewer 3 Report

This a very well written and exhaustive review on the topic of pancreatic cancer tumorigenesis. In my opinion the paper is acceptable in its present form 

Author Response

Thank you so much for your positive comments.